# Glycemic control in newly insulin-initiated patients with type 2 diabetes mellitus: A retrospective follow-up study at a university hospital in Ethiopia

**Ashenafi Kibret Sendekie** [1]*, **Achamyeleh Birhanu Teshale**[2], **Yonas Getaye Tefera**[1]

**1** Department of Clinical Pharmacy, School of Pharmacy, College of Medicine and Health Sciences, University of Gondar, Gondar, Ethiopia, **2** Department of Epidemiology and Biostatistics, Institute of Public health, College of Medicine and Health Sciences, University of Gondar, Gondar, Ethiopia

* ashukib02@yahoo.com, Ashenafi.kibret@uog.edu.et

## Abstract

### Background

Though many trials had examined the effectiveness of taking insulin with or without oral agents, there are limited real-world data, particularly among patients with type 2 diabetes mellitus (T2DM) in the resource limited settings. This study aimed to examine level of glycemic control among patients with T2DM after initiation of insulin and factors associated with poor glycemic control.

### Methods

An analysis of retrospective medical records of patients with T2DM who initiated insulin due to uncontrolled hyperglycemia by oral agents was conducted from 2015–2020 in the University of Gondar Comprehensive Specialized Hospital. Difference in median fasting plasma glucose (FPG) before and after insulin initiations was examined by a Wilcoxon signed-rank test. Kruskal Wallis test was performed to explore difference in the median level of FPG among treatment groups. A logistic regression model was also used to identify associated factors of poor glycemic control after insulin initiation. Statistical significance was declared at $p < 0.05$.

### Results

Of 424 enrolled patients with T2DM, 54.7% were males and the mean age was 59.3±9.3 years. A Wilcoxon signed-rank test showed that there was significant deference in FPG before and after insulin initiation ($P < 0.001$). A declining trend of blood glucose was observed during the 1-year follow-up period of post-initiation. However, majority of the participants did not achieve target glucose levels. Participants who had higher FPG and systolic blood pressure (SBP) before insulin initiation were found more likely to have poor glycemic control after insulin initiation. Similarly, patients who received atorvastatin compared with simvastatin were found to have poor glycemic control in the post-period of initiation ($P =$

**Funding:** Funding was received from the University of Gondar to conduct this study.

**Competing interests:** The authors have declared that no competing interests exist.

0.04). Premixed insulin was associated with a lower likelihood of poor glycemic control than neutral protamine Hagedorn (NPH) insulin (P < 0.001).

## Conclusion

Following insulin initiation, a significant change in glycemic level and declining trend of FPG was observed during a 1-year follow-up period. However, the majority of patients still had a poorly controlled glycemic level. Appropriate management focusing on predictors of glycemic control would be of a great benefit to achieve glycemic control.

## Introduction

Diabetes continues to be one of the most common non-communicable chronic diseases, and described by elevated blood glucose levels [1,2]. Type 2 Diabetes Mellitus (T2DM) is the main type of diabetes in adults, which is characterized by a gradual deterioration of glycemic control due to progressive pancreatic beta-cell dysfunction of insulin secretion on the background of increasing of insulin resistance [3–5]. In long term, uncontrolled hyperglycemia leads to complications of cardiovascular diseases (CVDs) and microvascular complications like damages of eyes, kidneys and nerves, and finally, leads for death [1]. In addition to these common macrovascular and microvascular complications, diabetes has been associated with another important complications like cochlear dysfunction [6], and sexual dysfunction and fracture [7,8].

The International Diabetes Federation (IDF) diabetes Atlas reported in 2021 that the prevalence of diabetes in adults was 10.5% (537 million) and estimated to be 12.2% (783 million) in 2045 worldwide, while in Africa it was 4.5% (24 million) in 2021 and projected to be 5.2% (55 million) in 2045. This demonstrates diabetes has become a major public health problem particularly in under developed countries with a significant social and financial implications [9]. In Ethiopia, the prevalence of diabetes was estimated as high as 6.5% [10]; and it makes one of the largest diabetes population in the sub-Saharan Africa.

The main goal of T2DM treatment is to safely achieve and maintain glycemic control to reduce risk of diabetes related microvascular and macrovascular complications, and in the long run, diabetes related mortality. With this regard, the American Diabetes Association (ADA) recommends glycemic targets of glycosylated hemoglobin (HbA1c) values to be <7% and fasting plasma glucose (FPG) levels of 70 to 130 mg/dl [11]. Even though patients with T2DM may initially attain glycemic control with oral antidiabetics (OADs), achieving a target glycemic level becomes increasingly difficult due to disease progression, and most patients ultimately require multidrug regimens and insulin initiation [12–14].

Evidences has shown that insulin therapy improves diabetes symptoms and delay of insulin initiation may lead to significant number of diabetes related complications [15,16]. Although timely initiation of insulin for T2DM has been recommended to prevent diabetes related complications by early establishment of strict glycemic control and pancreatic beta-cell protection [17], greater proportion of patients with suboptimal glucose level tend to delay insulin therapy [18] due to fear of hypoglycemia and weight gain [19,20]. In some case, patients may not take medications intentionally, driven by their emotions they may conceal it and become nonadherence to the recommended medication, which in turn lead to potential diabetes related complications and dire consequences [21], therefore, the need to educate patients about management practices and lifestyle modifications to achieve good treatment outcome could be mandatory [22]. Healthcare providers-patient relationship is also very crucial in the treatment

intensification and medication adherence. Moreover, the physician himself may also denote a risk factor for poor glycemic control due to the fear of potential drug's adverse effect and not providing appropriate patient's counseling [23,24]. On the other hand, the majority of patients with T2DM who initiate insulin therapy are also unable to achieve the target glycemic levels [25,26]. As a result, T2DM treatment guidelines have acknowledged a variety of factors can affect an individual's ability to reach the standardized glycemic goal and promote patient-centered management, and health service providers also request more and real-world data on which particular patient characteristics determines glycemic outcomes [27–29].

Though many studies had examined the effectiveness of taking insulin with or without oral agents [30–32], there are limited real-world data, particularly among patients with T2DM in resource limited-countries. To the best of our literature search, a single article that examine level of glycemic control in patients with T2DM after insulin initiation has not been published in low-income settings like Ethiopia, particularly in the study area. Identification of the factors associated with poor glycemic control by using data from routine clinical care settings and characterize the level of glycemic control is important. This will help to institute appropriate measure to improve glycemic control, and prevent long-term complications and organ damages related with diabetes [33]. Therefore, this study aimed to examine level of glycemic control in patients with T2DM after initiation of insulin and associated factors for poor glycemic control at the University of Gondar Comprehensive Specialized Hospital (UoGCSH), Northwest Ethiopia. This real-world data may help to understand the trends of glycemic control and factors associated with poor glycemic control in T2DM patients initiated with insulin in the resource limited settings.

## Methods and materials

### Study design and participants

Retrospective follow-up study was conducted from 2015 to 2020 using medical records of patients with T2DM at UoGCSH. Patients with age 18 years and above who initiated insulin due to inadequate glycemic control by OADs were recruited and then followed 1-year pre and post initiation. To be selected in the study, patients were required to be treated with insulin therapy during the indexed period of 2015–2020. The date that the first prescription with insulin identified was taken as index date. Patients should have available data for 1 year before and after the index date, received OADs before the index date and insulin after the index date. Patients were excluded in the study if they received a diagnosis of gestational diabetes or type I diabetes over the indexed period. Patients with incomplete medical records were also excluded from the study.

### Sampling size determination and sampling technique

The sample size was determined by using a single population proportion formula:

n = $Z^2$ p (1-p)/W, Where, n = sample size required, W = marginal error of 5% (w = 0.05), Z = the degree of accuracy required (95% level of significance = 1.96), P = the proportion of poor glycemic control in patients with T2DM treated with insulin-based therapy assumed to be 0.5(50%), this is because no appropriate prior study was conducted in the study setting and other areas with similar population background. Considering the possible incomplete patient records to be 10%, 424 patient records were enrolled in the final study and selected using systematic random sampling technique from the list of all eligible study population. Simple random sampling technique using lottery method was also used to select the first participant to be the starting point. Then using the sampling interval with coding of their medical records, participants were enrolled until the required sample size was fulfilled.

## Operational definitions

**Macrovascular complications:** diabetes associated complications related to cardiovascular outcomes such as stroke, ischemic heart disease, heart failure, coronary artery disease, peripheral vascular disease.

**Microvascular complications:** diabetes associated complications related to kidney (diabetic nephropathy), nerves (peripheral neuropathy) and eye problems (diabetic retinopathy)

**Renal problems:** include comorbidities diagnosed as acute and/or chronic kidney disease on the patients' physical medical records

**Data collection instruments and procedures.** The data extraction tools were prepared by reviewing different literatures and amendments were made considering the setting and nature of patient medical records. The data collection tool had four parts. The first consisted socio-demographic characteristics of patients and the rest were clinical characteristics and medications before insulin initiation, during initiation and after initiation. A socio-demographic characteristic of the patients includes; age, gender, residency, and duration of T2DM since diagnosis, duration of treatment with OADs. Whereas clinical characteristics includes laboratory results, physicians and nurse notes, prescribed and dispensed medications, diagnosis and procedures other details of patients visit to the hospital during the follow-up periods. Medications also includes types of diabetic medications, antihypertensive agents, lipid-lowering agents and other medications used for the treatment of presented comorbidities and complications of patients. A 2-year data (1 year before and 1 year after the indexed date) were recorded every three months from stored physical medical records of the patients, and printed laboratory results were also checked for some laboratory tests like FPG, serum creatinine (Scr.) and lipid profiles such as total cholesterol (TCL) and triglycerides (TG). Metformin with or without glibenclamide was used in in pre-period of initiation and then insulin (NPH or premixed) was initiated. Treatment intensification including dosing titration and frequency were modified based on the ADA recommendations.

## Data quality management and statistical analysis

The data collectors and supervisor were trained before the actual data collection. Pretest was done on 10% of sample size and some amendments was done. Once the medical record identification numbers were entered to the Microsoft excel 2016 and checked for repetition, the data was extracted. The supervisor has explicitly followed the data collection closely. Both the data collectors and supervisor checked the data for its completeness and missing information at each point before analysis. After checking the data completeness and cleanness, then coded and entered to Epi Info version 7 and exported to SPSS Version 26 for analysis.

Descriptive statistics such as frequencies and percentage were used for categorical variables and mean with standard deviation were used for continuous variables. Non-normality of the data for FPG, systolic blood pressure (SBP), diastolic blood pressure (DBP), lipid profiles and SCr. was examined by Q-Q plot and histogram, and median with an inter-quartile range (IQR) was used to measure their levels.

A Wilcoxon signed-rank test was used to examine the median score difference between paired FPG after and before insulin initiation. Median score difference in FPG between treatment groups (insulin alone Vs insulin plus metformin Vs insulin plus metformin plus glibenclamide) was explored by Kruskal-Wallis test. The Post-hoc test using a Pairwise Multiple-comparative analysis was also used to compare the glycemic difference between all paired treatment groups. The logistic regression model was fitted to assess variables associated with poor glycemic control after insulin-initiation. Variables, with $P \leq 0.25$ in the bivariable analysis, were entered for multivariable logistic regression analysis. Finally, the adjusted odds ratio

(AOR) with 95% confidence interval (CI) was reported, and a P-value < 0.05 was statistically significant.

### Glycemic outcome measurements

The glycemic outcome following insulin initiation in this study was examined by using level of FPG due to non-availability of HbA1c. The American diabetes association categorized glycemic control as good glycemic control: FPG levels of 70 to 130 mg/dl and poor glycemic control: FPG level of either <70mg/dl or FPG >130 mg/dl [11].

### Ethical considerations

The study was approved by the ethics approval committee of the University of Gondar with reference number of Sop/037/2020. The need for informed consent was waived by the ethics committee of the University of Gondar because the study did not directly involve the patients. Privacy and confidentiality were kept, and all methods were carried out in accordance with relevant guidelines and regulations.

## Results

### Socio-demographic and baseline clinical characteristics

From a total of 937 eligible patients with T2DM, 424 study subjects were included in the study. More than half of the analyzed subjects were male (54.7% and urban residents (59.9%). Most of study participants had hypertension (67%) along with diabetes, and patients were most commonly received metformin plus glibenclamide combination therapy prior to insulin initiation. Enalapril (59%) was also the most frequently prescribed cardiovascular medication. The median (IQR) of FPG level at the index date was 350 (179–401) mg/dl, Tables 1 and 2.

### Patterns of medication and level of glucose during the follow-up period

The majority of patients had received a combination of metformin and glibenclamide therapy in all follow-up periods prior to insulin initiation, and nearly more than two-thirds of patients were received a combination of insulin and metformin during post-initiation periods (S1 Fig). Furthermore, the median FPG level was lower among patients treated by metformin than patients treated by a combination of metformin plus glibenclamide in all follow-up periods. Similarly, the median FPG level was also lower among patients received triple therapy (insulin plus metformin plus glibenclamide) compared with patients on dual therapy of insulin plus metformin and insulin single therapy in the post- initiation periods (S2 Fig). Among the types of insulins, significant number of patients had received NPH insulin-based therapy. However, frequency of clinical hypoglycemia was recorded more among patients treated with premixed insulin-based regimen than patients treated by NPH insulin-based therapy.

### Glycemic control following insulin initiation and trends of glucose level

The level of blood glucose was compared before and after insulin initiation. On average, the study participants had worse before (Mdn = 350) than after insulin initiation (Mdn = 175.5) at the 3$^{rd}$ month of post-initiation period, and gradual declining of FPG level in a 1-year follow-up period was also observed. A Wilcoxon signed-rank test indicated that this difference was statistically significantly, T = 90,100, Z = -17.84, P < 0.001. However, significant number patients did not achieve a target glycemic level after insulin initiation, three-fourths (75%) and 61.3% of the study participants did not achieve the target FPG level at 3$^{rd}$ and 12$^{th}$ month of post-initiation periods, respectively (Fig 1). The study participants achieved target blood

**Table 1. Socio-demographic and baseline clinical characteristics of newly insulin-initiated patients with T2DM having follow-up at UoGCSH from 2015–2020 (N = 424).**

| Characteristics | | Frequency (%) | Mean ± SD or Median (IQR) |
|---|---|---|---|
| Sex | Male | 232 (54.7) | |
| | Female | 192 (45.3) | |
| Age (years) | Mean ± SD | - | 59.3± 9.3 |
| Weight (Kg) | Mean ± SD | - | 65.7± 8.2 |
| Residency | Urban | 254 (59.9) | |
| | Rural | 17 (40.1) | |
| Clinical characteristics | | | |
| Years since T2DM diagnosis. | Mean ±SD | - | 13.4±4.0 |
| Years since OADs started | Mean ±SD | - | 12.9± 3.8 |
| Comorbidities and Complications | Hypertension | 284 (67.0) | |
| | Dyslipidemia | 151 (35.6) | |
| | Macrovascular Complications | 66 (15.6) | |
| | Bacterial infection | 27 (6.4) | |
| | Microvascular Complications | 25 (5.9) | |
| | Diabetic Keto-acidosis (DKA) | 22 (5.2) | |
| | Renal problems (AKI and CKD) | 15 (3.5) | |
| | Retroviral infection | 12 (2.8) | |
| | Bronchial asthma | 6 (1.4) | |
| | Thyrotoxicosis | 5 (1.2) | |
| Laboratory Parameters | | | |
| | FPG (mg/dl) at 12th month before the index date | - | 188(166–209) |
| | FPG (mg/dl) at the index date | - | 350(179–401) |
| | SBP (mmHG) at 12th month before the index date | - | 130(130–140) |
| | DBP (mmHG) at 12th month before the index date | - | 70(70–80) |
| | SBP (mmHG) at the index date | - | 140(130–140) |
| | DBP (mmHG) at the index date | - | 80.00(71.25–90) |
| | Creatinine (mg/dl) at 12th month before the index date | - | 0.88(0.81–1.13) |
| | Creatinine (mg/dl) at the index date | - | 0.89(0.81–1.06) |
| | Total cholesterol(mg/dl) at 12th month before the index date | - | 178.12(125.5–196.25) |
| | Total cholesterol at the index date | - | 179(165.755–216.75) |
| | Total triglyceride (mg/dl) at 12th month before the index date | - | 161(140.01–210) |
| | Total triglyceride (mg/dl) at the index date | - | 154(140.25–190.75) |

AKI, Acute kidney injury; CKD, Chronic kidney disease; SD, Standard deviation; IQR, Inter quartile range.

glucose level with an average time of 6.7±3.4 months during the 1–year follow-up period after insulin initiation. As shown in the Fig 2, the level of FPG was increased since 12th month of pre-period of initiation until the index date but a sharp decreasing in FPG initially followed by gradual decline was observed through a one-year follow-up period in the post-insulin initiation time.

## Difference in blood glucose between treatment groups after insulin initiation

A significantly reduced level of FPG after insulin initiation was recorded with the overall median (IQR) score of 175.5 (135–209) mg/dl at the 3rd month of post-initiation period.

**Table 2. Distribution of baseline medications used to treat study participants (N = 424).**

| Medications | | | Frequency | Percent |
|---|---|---|---|---|
| OADs | Before insulin initiation | Metformin | 62 | 14.6 |
| | | Metformin plus Glibenclamide | 362 | 85.4 |
| | During insulin initiation | Metformin | 56 | 13.2 |
| | | Metformin plus Glibenclamide | 368 | 86.8 |
| Antihypertensive agents | Enalapril | | 251 | 59.0 |
| | Amlodipine | | 25 | 5.9 |
| | Hydrochlorothiazide | | 77 | 18.2 |
| | Atenolol | | 15 | 5.3 |
| | Metoprolol | | 12 | 2.8 |
| | Nifedipine | | 14 | 3.3 |
| | Furosemide | | 4 | 0.9 |
| Lipid lowering agent | Atorvastatin | | 103 | 24.3 |
| | Simvastatin | | 79 | 18.6 |
| Aspirin (ASA) | | | 61 | 14.4 |
| Amitriptyline | | | 19 | 4.5 |
| Antibiotics | Ceftriaxone | | 23 | 5.4 |
| | Metronidazole | | 7 | 1.7 |
| | Vancomycin | | 4 | 0.9 |
| Gastrointestinal | Omeprazole | | 14 | 3.3 |
| Anti-Retroviral Therapy | TDF/3TC/DTG | | 10 | 2.4 |
| | AZT/3TC/DTG | | 2 | 0.5 |
| Anti-asthmatic | Salbutamol | | 6 | 1.4 |
| | Beclomethasone | | 6 | 1.4 |
| Anticoagulant | Warfarin | | 6 | 1.4 |
| Anti-thyroid | Propyl thiouracil | | 5 | 1.2 |

TDF, Tenofovir disoproxil fumarate; 3TC, Lamivudine; DTG, Dolutegravir; AZT, Zidovudine.

However, patients who were treated by triple therapy of insulin plus metformin plus glibenclamide had worse glycemic level (Mdn = 200) than patients treated by combination therapy of insulin plus metformin (Mdn = 170) and insulin alone (Mdn = 170.5). A Kruskal-Walli's test revealed that the deference in level of FPG among treatment groups was statistically significant, H (2) = 19.51, P < 0.001. The Post-hoc tests using a Pairwise Multiple-comparative analysis showed that there was a statistically significant difference in level of FPG between a combination therapy of insulin plus metformin (Mdn = 170) vs insulin plus metformin plus glibenclamide triple treatment groups (Mdn = 200), P < of 0.001. There was also a difference in proportion of patients achieving glycemic control among these treatment groups. One-third of patients (33.3%) from insulin treated group, 29.8% from insulin plus metformin and 15.2% from insulin plus metformin plus glibenclamide treatment groups achieved the target FPG level.

However, significant difference in level of FPG among treatment groups was not observed at the 12th month of post-initiation period, and the overall median (IQR) of FPG was 139 (114–159.75). A Kruskal-Wallis test also showed that the difference in level of median FPG among treatment groups was not statistically significant, H (2) = 3.27, P = 0.195. Nearly two–fifths of patients had achieved target FPG level among all treatment groups, 40.7% from insulin, 38.2% from insulin plus metformin and 38% from insulin plus metformin plus glibenclamide treatment groups.

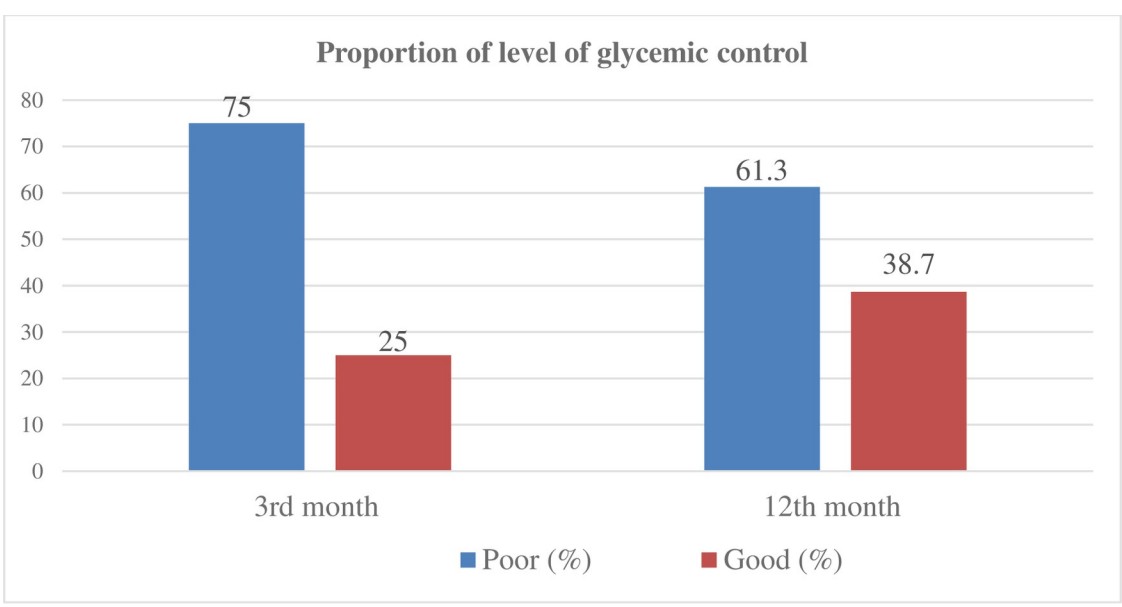

**Fig 1. Proportion of participants to level of glycemic control after insulin initiation (N = 424).**

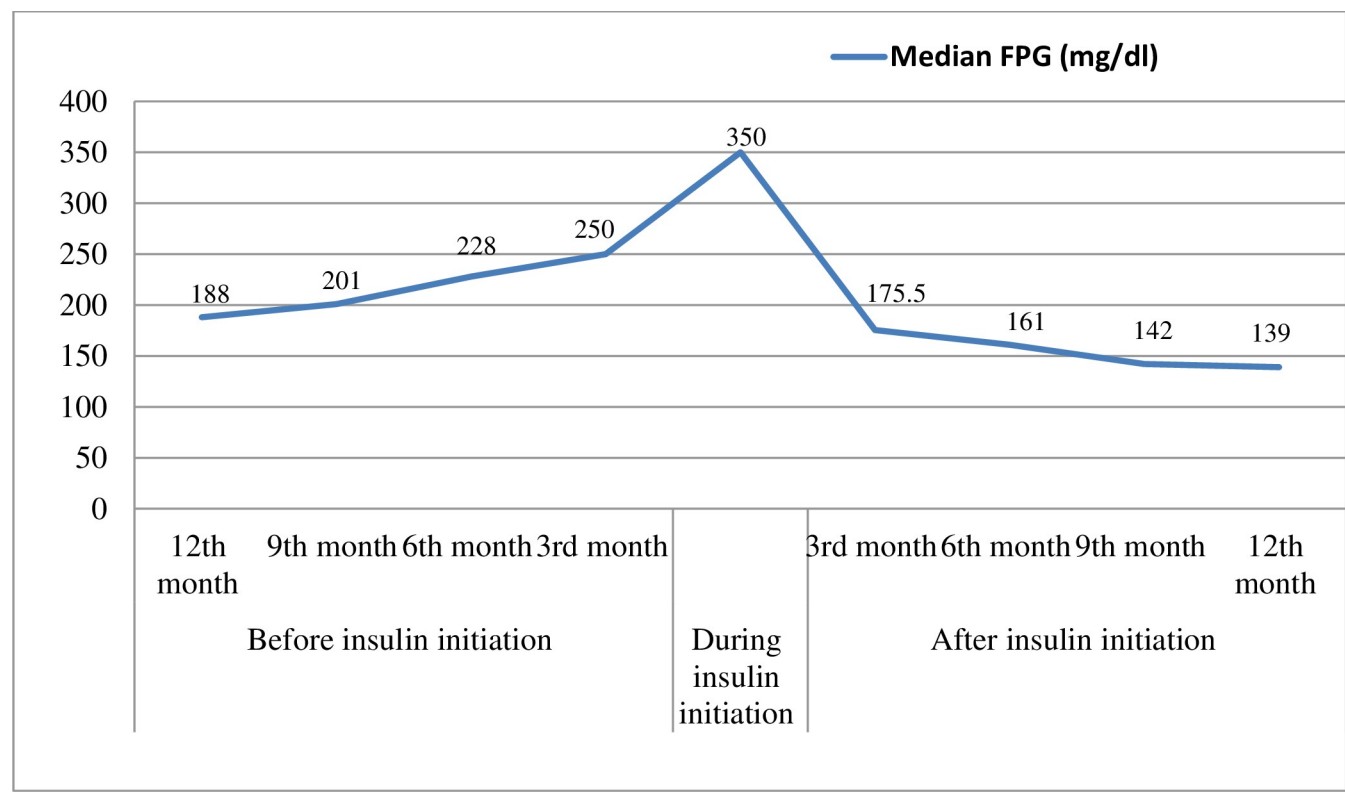

**Fig 2. Trend of fasting blood glucose levels of participants during the 2-years of follow-up periods.**

## Determinants of glycemic control after insulin initiation

The multivariable logistics regression model showed an association between post-initiation period of poor glycemic control and higher FPG and SBP levels during the index date, and use of atorvastatin compared to simvastatin was also associated with poor glycemic control in the post period of insulin initiation. Higher FPG and SBP levels during insulin initiation were significantly associated with poor glycemic control after 3rd month of insulin initiation, [AOR: 1.018(1.009–1.028); P < 0.001] and [AOR: 1.074(1.028–1.127); P = 0.004], respectively. Similarly, patients who were treated with atorvastatin were found more likely to have poor glycemic control than patients who were treated by simvastatin at the 3rd month of post-initiation, [AOR: 2.573 (1.046–6.328); P = 0.04]. On the other hand, premixed insulin was associated with a lower likelihood of poor glycemic control as compared with NPH insulin, [AOR: 0.147 (0.056–0.368); P ≤ 0.001], Table 3.

## Discussion

This is an institutional based retrospective follow-up study focused on examining glycemic control in insulin-initiated patients with T2DM due to inadequate glycemic control by OADs alone. The results highlight that level of glycemic control differ meaningfully from more strictly controlled trials [34,35]. Thus, in order to identify which specific patient factors affect glycemic outcomes, generating data from a real-world clinical setting was as such important.

The current study revealed that the initiation of insulin in patients with T2DM resulted in a significant decreasing level of glucose after insulin initiation. Consistent to the previous studies assessing glycemic control in patients with T2DM after initiation of insulin therapy [25,26,36–39], the current result showed that a lower proportion of patients achieved target FPG level during the 1-year follow-up period. This indicate that the current target blood glucose goal may be unachievable for many patients with T2DM even after insulin initiation. However, this result is inconsistent with findings from clinical trials [34,35]; a significant number of patients achieving glycemic control. But it would be recognized that these clinical trials may not indicate the real life of clinical care due to nature of its' design with treat to target trail, narrow inclusion criteria, close monitoring and regular follow-up during the study. Furthermore, it would be noted that vast majority of patients in the current study did not have a regular HbA1c test as ADA recommendations. Consequently, this used FPG to examine glycemic control, and it might have different result as compared with clinical trials using HbA1c. Therefore, to obtain better glycemic outcome in the real-world clinical settings, insulin intensification and titration could be based on the actual specific patients' characteristics which potentially affect glycemic control. However, the treatment might not have been intensified and titrated effectively because of fear of adverse effects like hypoglycemia, and the need to educate patients about insulin administration and adverse effects would be mandatory. Patient educational on lifestyle modification and management practices could be also an important component to achieve better treatment outcome [22], and as a result patients and healthcare providers would give an equal attention to patient education as equal as medication management.

Consistent with previous studies [40–42], the blood glucose level at 3rd month of insulin initiation was significantly different among treatment groups. The results may be explained by patients who have worse glycemic level may require a combination of oral medications besides insulin to achieve their target glucose levels. The current study also disclosed that initial oral medications were continued or added in the regimen for patients with worse glycemic level following insulin initiation. In contrast, nearly equivalent glycemic levels were achieved at the 12th month of insulin initiation in all treatment groups, in consistent with the previous study [43]. This might be achieved because of increased treatment titration and treatment

**Table 3. Association of variables with poor glycemic control after insulin initiation.**

| Variables | | Glycemic control | | COR (95% CI) | P-value | AOR (95% CI) | P-value |
|---|---|---|---|---|---|---|---|
| | | Poor | Good | | | | |
| Duration in years since T2DM diagnosis (mean ± SD) | | 13.6±4 | 12.8±3.8 | 1.052 (0.993–1.114) | 0.086 | 0.567 (0.306–1.048) | 0.07 |
| Duration in years since OADs initiation (mean ± SD) | | 13.1±3.9 | 12.2±3.5 | 1.062 (1.000–1.127) | 0.049 | 1.777 (0.940–3.356) | 0.077 |
| FPG at the index date (Median (IQR) | | 365 (326–406) | 323 (306–349) | 1.014 (1.009–1.018) | 0.000 | 1.018 (1.009–1.028) | 0.000* |
| SBP at the index date (Median (IQR) | | 140 (130–140) | 130 (130–140) | 1.037 (1.014–1.061) | 0.002 | 1.074 (1.028–1.127) | 0.004* |
| Residency | Urban Rural | 180 138 | 74 32 | 0.564 (0.352–0.903) 1 | 0.017 | 0.586 (0.225–1.525) 1 | 0.273 |
| OADs during insulin initiation | Metformin plus glibenclamide Metformin | 280 38 | 88 18 | 1.507 (0.819–2.773) 1 | 0.187 | 0.724 (0.218–2.046) 1 | 0.598 |
| Hypertension | Yes No | 233 85 | 71 35 | 1.351 (0.840–2.173 1 | 0.214 | 0.517 (0.172–1.556) 1 | 0.241 |
| Furosemide after insulin initiation | Yes No | 1 317 | 6 100 | 0.053 (0.006–0.442) 1 | 0.007 | 0.157 (0.015–1663) 1 | 0.124 |
| Lipid lowering agents | Atorvastatin Simvastatin | 118 32 | 30 19 | 2.335 (1.166–4.679) 1 | 0.017 | 2.573 (1.046–6.328) 1 | 0.04* |
| ASA after insulin initiation | Yes No | 49 269 | 22 84 | 0.696 (0.397–1.217) 1 | 0.203 | 1.160 (0.469–2.867) 1 | 0.748 |
| Diabetes medications | Insulin plus metformin Insulin plus metformin plus glibenclamide Insulin | 179 126 16 | 78 22 8 | 1.187 (0.487–2.891) 2.696 (1.034–7.028) 1 | 0.006 0.706 0.043 | 0.770 (0.125–4.741) 0.568 (0.076–4.219) 1 | 0.801 0.778 0.58 |
| Type of insulin | Premixed NPH | 41 277 | 45 61 | 0.201 (0.121–0.333) 1 | 000 | 0.147 (0.059–0.368) 1 | 0.000* |

ASA, Aspirin; COR, Crude odds ratio; AOR, Adjusted odds ratio; IQR, inter-quartile range; P-value * indicates the statistically significant variables at P < 0.05.

modifications for patients those with poor glycemic level in the early period of insulin initiation. However, regardless of differences in the level of blood glucose throughout the follow-up periods, improved change in glycemic levels after insulin initiation was observed in all

treatment groups. This is in agreement with previous studies conducted across the globe [36,44–46], which shows significant change in blood glucose level after insulin initiation in patients with T2DM who were initially treated with OADs. Moreover, this study also showed that during a one-year follow-up period of post-insulin initiation, a continual declining in blood glucose level was observed from the insulin initiation to the end of follow-up period.

The current study also demonstrated about factors affecting glycemic control in insulin-initiated patients with T2DM. Similar to other studies [47–50], the current finding showed that higher baseline blood glucose level was significantly associated with poor glycemic outcome in post-initiation period. This suggests that patients with good baseline glycemic control have minimal deterioration of glycemic level after insulin initiation and the probability of achieving the target glycemic level may strongly associated with the baseline level of glucose. Thus, early insulin initiation in patients having indication might be important to achieve the target blood glucose goals and to prevent the deterioration by early establishment of strict glycemic control. The strict glycemic control also activates anti-inflammatory, anti-apoptotic and anti-oxidative stress mechanisms, as well as increases endothelium protection, reduces free fatty acid, presents an anti-glucotoxic effect, and also improves both insulin resistance and cardiac fuel metabolisms, which are vital in pancreatic beta-cell protection and reducing of complications onset. All these mechanisms are involved in pancreatic beta-cell preservation and reduced onset of complications [51,52]. In this study, the poor glycemic control following insulin initiation was also significantly increased with a clinically relevant unit increasing of SBP before insulin initiation. This indicates that the target blood glucose level may be difficult to achieve in patients with higher blood pressure even with insulin initiation. This finding might explain that uncontrolled blood pressure could result in poor glycemic control because patients with higher blood pressure sustain a resistance to insulin which decreases insulin uptake and altering delivery of insulin and finally result in impaired glucose uptake [53]. Blood pressure control is so important in patients with T2DM to curb the worse progress of glycemic level.

In the current study, patients who were treated by atorvastatin were found more likely to have poor glycemic control compared with patients treated with simvastatin following insulin initiation. This is consistent with previous studies, which demonstrated that high intensity dose of atorvastatin was associated with the worsening and deterioration of glycemic level compared with moderate intensity statins [54–56]. The finding may prove that statin treatment has a role of downregulation of glucose transporter in adipocytes, which may result in insulin resistance and glycemic deterioration in patients with diabetes especially with high intensity statin therapy. Another study showed that there is no significant changes in glycemic level between atorvastatin and other treatment groups [57], but it was a study with very small number of study participants and unknown dose of atorvastatin used; the average dose of atorvastatin in the current study was in the range of high intensity with 40 mg/day. Moreover, in consistent with the previous study [58], the finding from the current study revealed that premixed insulin-based regimen was found significantly associated with a lower likelihood of poor glycemic control compared with NPH insulin-based regimen. The finding may suggest that patients treated with premixed insulin-based regimen may have a better glycemic outcome than patients treated with NPH insulin. It might be because of that the premixed inulin has two types of insulin in the preparation which can be important to adjust both the postprandial and the basal blood glucose levels. However, frequent episode of hypoglycemia was observed in patients treated with the premixed insulin-based therapy compared with patients treated by NPH insulin-based therapy. Thus, frequent and close monitoring of hypoglycemia is required when patients initiated with premixed insulin-based therapy. Generally, in patients with type 2 diabetes glycemic control needs multifactorial interventions and appropriate management which have been proved to be vital not only to optimize a good glycemic profile, but

also to reduce complications onset, in particular cardiovascular ones, and should represent the gold standard for this subset of patients' treatment [59].

Our study has strengths and some limitations. The study is the first to explore glycemic control and determinants in newly insulin-initiated patients with T2DM who failed to achieve glycemic control by OADs in the study area. It may be used as a benchmark for clinicians and future researchers to examine glycemic control and predictors in post- insulin initiations further with prospective and larger populations. This retrospective study was conducted in preexisting patients' medical records and some variables like AKI and CKD, macro and micro complications may not be consistent throughout the patients' physical medical records. Besides, HbA1c, which reflects the average blood glucose level over the past three months, was not used because of non-availability. Instead, fasting plasma glucose, which shows a short-term glycemic index, was used to determine glycemic control.

## Conclusion

The initiation of insulin to the therapeutic regimen of insulin naive T2DM patients brought a significant change in glycemic level and a declining trend of FPG during a 1year post-initiation follow-up period. However, a significant proportion of patients had poor level of glycemic control even after insulin initiation. Patients who had higher level of FPG and SBP before insulin initiation, and patients treated with atorvastatin were found more likely to have poor glycemic control in the post-initiation period. Similarly, premixed insulin-based therapy was associated with a lower likelihood of poor glycemic control as compared to NPH insulin-based therapy. Therefore, appropriate management of patients focusing on independent predictors of glycemic control would be of a great benefit to achieve glycemic target.

## Supporting information

**S1 Fig. Proportion of the study subjects with respective medications during the 2-years follow-up period (N = 424).**
(TIF)

**S2 Fig. Fasting blood glucose level with respective medications during the 2-years follow-up period.**
(TIF)

**S1 File. Dataset.**
(SAV)

## Acknowledgments

The authors would like to thank the hospital nurses and pharmacists for collecting the data and hospital medical record unit managers for their assistance during data abstraction.

## Author Contributions

**Conceptualization:** Ashenafi Kibret Sendekie.

**Data curation:** Ashenafi Kibret Sendekie, Achamyeleh Birhanu Teshale, Yonas Getaye Tefera.

**Formal analysis:** Ashenafi Kibret Sendekie, Achamyeleh Birhanu Teshale, Yonas Getaye Tefera.

**Investigation:** Ashenafi Kibret Sendekie.

**Methodology:** Ashenafi Kibret Sendekie, Achamyeleh Birhanu Teshale, Yonas Getaye Tefera.

**Project administration:** Ashenafi Kibret Sendekie.

**Resources:** Ashenafi Kibret Sendekie.

**Supervision:** Achamyeleh Birhanu Teshale, Yonas Getaye Tefera.

**Writing – original draft:** Ashenafi Kibret Sendekie.

**Writing – review & editing:** Achamyeleh Birhanu Teshale, Yonas Getaye Tefera.

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
