## [Decision Letter · Decision Letter 0]

20 Apr 2022

PONE-D-22-08211Glycemic control in patients with type 2 diabetes who initiate insulin: a retrospective follow-up study at a university hospital in EthiopiaPLOS ONE

Dear Dr. Sendekie,

Thank you for submitting your manuscript to PLOS ONE. After careful consideration, we feel that it has merit but does not fully meet PLOS ONE’s publication criteria as it currently stands. Therefore, we invite you to submit a revised version of the manuscript that addresses the points raised during the review process.

We look forward to receiving your revised manuscript.

Kind regards,

Ferdinando Carlo Sasso, PhD, MD

Academic Editor

PLOS ONE

Journal Requirements:

Additional Editor Comments:

Three external peer reviewers carefully reviewed the manuscript. In this version, the paper cannot be accepted for publication in Plos One. The main issue concerns the originality of the manuscript.

If the authors are able to adequately address the issues raised by the reviewers they can resubmit a revised version of the manuscript.

Reviewers' comments:

Reviewer's Responses to Questions

**Comments to the Author**

1. Is the manuscript technically sound, and do the data support the conclusions?

Reviewer #1: Yes

Reviewer #2: Yes

Reviewer #3: No

2. Has the statistical analysis been performed appropriately and rigorously? 

Reviewer #1: Yes

Reviewer #2: Yes

Reviewer #3: I Don't Know

3. Have the authors made all data underlying the findings in their manuscript fully available?

Reviewer #1: Yes

Reviewer #2: Yes

Reviewer #3: No

4. Is the manuscript presented in an intelligible fashion and written in standard English?

Reviewer #1: Yes

Reviewer #2: Yes

Reviewer #3: No

5. Review Comments to the Author

Reviewer #1: The paper “Glycemic control in patients with type 2 diabetes who initiate insulin: a retrospective follow-up study at a university hospital in Ethiopia" by Sendekie et al. deals with a very important topic and provides data of high clinical interest, especially for low-income countries.

The article is well written and only minor spell check is necessary. The paper has a good design. The article is logically divided into sections and subsections. The work has a good degree of novelty and of good interest to the readers. The introduction and discussion should be revised.

Comments:

1) Introduction, line 7: Type 2 diabetes may be accompanied by several complications as the author have stated. In my opinion it should also be reported another important and not well-known complication, which is the cochlear dysfunction (doi: 10.1016/s0026-0495(99)90141-5).

2) Introduction, line 24-25: patient’s nonadherence to pharmacological treatment is often patients’ intentional choice, and, driven by their emotions they may conceal it, which may in turn lead to potentially dire consequences (doi: 10.26355/eurrev_202202_28093). Doctor-patient relationship is fundamental in drug adherence. Moreover, the physician himself may also represent a risk factor for low glycaemic control due to the fear of potential drug’s adverse effect and not appropriate patient’s education (doi: 10.1097/01.hjh.0000239303.93872.31; doi: 10.1001/jama.2019.11489).

3) Discussion: the authors have reported an improved glycaemic control in patient with a better blood pressure monitoring and statin use. The role of multifactorial intervention in type 2 diabetes has been proved to be fundamental not only to reach a good glycaemic profile, but also to reduce complications onset, in particular cardiovascular ones, and should represent the gold standard for this subset of patients’ treatment (doi: 10.1186/s12933-021-01343-1).

4) Discussion, page 14, line 16-18: the strict glycaemic control also activates anti-inflammatory, anti-apoptotic and anti-oxidative stress mechanisms, as well as increases endothelium protection, reduces free fatty acid, presents an anti-glucotoxic effect, and also improves both insulin resistance and cardiac fuel metabolisms. All these mechanisms are involved in pancreatic beta-cell protection and reduced complications onset (doi: 10.1016/j.diabres.2021.108959; doi: 10.1155/2018/3106056)

Reviewer #2: This is an interesting update about glycemic control in patients with type 2 diabetes who initiate insulin in Ethiopia. Anyway, I have some comments and suggestions for the authors.

Abstract

- see better acronym for Type 2 diabetes.

Introduction

- please, about complications includes erectile dysfunction and fracture (see these ref: doi: 10.1002/dmrr.3494; doi: 10.2337/dc18-1965).

- please, update your ref about IDF Atlas ADA standard with the latest.

- after ref 15,16, about the need to educate patients, you can use ref DOI: 10.1007/s42000-018-0005-9.

Material and methods

- have you used any glucometro in particular?

Results

- have you data about HbA1c?

- have you data about the different type of anti hyperglycemic drugs used?

Discussion

- about appropriate management, could be important to emphasize thew role of educational therapy (see ref. DOI: 10.1007/s42000-018-0005-9)

Reviewer #3: In the current article the authors investigated the effectiveness of taking insulin with or without oral agents among patients with type 2 diabetes mellitus (T2DM) in the resource limited-countries like Ethiopia. Thus, in analysis of retrospective medical records of patients with T2DM who initiated insulin due to uncontrolled hyperglycemia of 424 enrolled patients with T2DM, there was significant deference in FPG before and after insulin initiation (P < 0.001), and a declining trend of blood glucose during a 1-year follow-up period of post-initiation was observed. However, majority of the participants did not achieve target glucose levels. Participants who had higher FPG and systolic blood pressure (SBP) before insulin initiation were found more likely to have poor glycemic control after insulin initiation. Similarly, patients who received atorvastatin than simvastatin were found to have poor glycemic control in the post-period of initiation (P =0.04), whereas premixed insulin was associated with a lower likelihood of poor glycemic control than neutral protamine Hagedorn (NPH) insulin (P < 0.001).

Therefore, they concluded that following insulin initiation, a significant change in glycemic level and declining trend of FPG during a 1-year post-initiation period was established. However, the majority of patients were under poor level of glycemic control. Appropriate management focusing on predictors of glycemic control would be of great benefit to achieve glycemic control.

The article looks as other just published articles on the same theme. However, my question is: what is new from the current article? What is the current article adding to the current literature on the topic of glycemic control in the diagnosis and treatment of the patients with type diabetes? It is well known that the insulin therapy could be used with higher success rate to achieve the best glycemic control in patients with type 2 diabetes. In my opinion we cannot publish this article.

6. PLOS authors have the option to publish the peer review history of their article (what does this mean?). If published, this will include your full peer review and any attached files.

Reviewer #1: No

Reviewer #2: No

Reviewer #3: No

---

## [Author Response · Author response to Decision Letter 0]

2 May 2022

Responses to the review’s comments

Dear PLOS ONE Academic Editor

We would like to thank you for your crucial comments on our paper (Manuscript Number: PONE-D-22-08211). We have tried to address all the suggested comments, which we believe it could strengthen our paper. We hope this render our paper to be considered for publication in your reputed journal. 

We, the authors would like to let you know that we made our best changes given by both reviewers and the editor point by points to the raised comments and recommendations. In addition, throughout our revision we made some corrections too. All the manuscript changes notified using tracking changes. 

Comments from the editor:

1#....Journal requirements:

Please ensure that your manuscript meets PLOS ONE's style requirements, including those for file naming

Author reply: Thank you for your recommendations to assure adherence to the Manuscript Template requirements of the journal. Considering to your recommendation, we have adjusted it accordingly. 

2#.....Additional Editor Comments:

Three external peer reviewers carefully reviewed the manuscript. In this version, the paper cannot be accepted for publication in Plos One. The main issue concerns the originality of the manuscript.

If the authors are able to adequately address the issues raised by the reviewers they can resubmit a revised version of the manuscript.

Author reply: Thank you very much for your recommendations to improve the whole manuscript of our work. With reminding the given recommendations and comments we rewrite the corrected parts with point-by-point and indicated with track changes. 

Response to Reviewers’ comments:

Reviewer #1

The paper “Glycemic control in patients with type 2 diabetes who initiate insulin: a retrospective follow-up study at a university hospital in Ethiopia" by Sendekie et al. deals with a very important topic and provides data of high clinical interest, especially for low-income countries.

The article is well written and only minor spell check is necessary. The paper has a good design. The article is logically divided into sections and subsections. The work has a good degree of novelty and of good interest to the readers. The introduction and discussion should be revised.

Comments:

Introduction

1#... Introduction, line 7: Type 2 diabetes may be accompanied by several complications as the author have stated. In my opinion it should also be reported another important and not well-known complication, which is the cochlear dysfunction (doi: 10.1016/s0026-0495(99)90141-5).

Author reply: We would like say thank you to the suggestion and the recommendations on this section to add important points. We actually accepted your suggestions and recommendations and it has been shown with track changes in the manuscript (page 3, lines 8-10).

2#... Introduction, line 24-25: patient’s nonadherence to pharmacological treatment is often patients’ intentional choice, and, driven by their emotions they may conceal it, which may in turn lead to potentially dire consequences (doi:10.26355/eurrev_202202_28093). Doctor-patient relationship is fundamental in drug adherence. Moreover, the physician himself may also represent a risk factor for low glycemic control due to the fear of potential drug’s adverse effect and not appropriate patient’s education (doi:10.1097/01.hjh.0000239303.93872.31;doi: 10.1001/jama.2019.11489). 

Author reply: Thank you for the given suggestions and based on the raised points we had corrected and revised the manuscript accordingly (page 4, lines 1-9). 

Discussion 

#3… Discussion: the authors have reported an improved glycemic control in patient with a better blood pressure monitoring and statin use. The role of multifactorial intervention in type 2 diabetes has been proved to be fundamental not only to reach a good glycemic profile, but also to reduce complications onset, in particular cardiovascular ones, and should represent the gold standard for this subset of patients’ treatment (doi: 10.1186/s12933-021-01343-1).

Author reply: Thank you, such evidences will improve the manuscript quality further. Regarding this we included the recommended evidence in the revised manuscript (page 15, lines 23-26).

#4…Discussion, page 14, line 16-18: the strict glycaemic control also activates anti-inflammatory, anti-apoptotic and anti-oxidative stress mechanisms, as well as increases endothelium protection, reduces free fatty acid, presents an anti-glucotoxic effect, and also improves both insulin resistance and cardiac fuel metabolisms. All these mechanisms are involved in pancreatic beta-cell protection and reduced complications onset (doi: 10.1016/j.diabres.2021.108959; doi: 10.1155/2018/3106056).

Authors reply: We are grateful for the important evidence which can strength the discussion. Thus, we included the recommended evidence and indicated with track changes in the main document (page 14, lines 21-26). 

Reviewer #2

This is an interesting update about glycemic control in patients with type 2 diabetes who initiate insulin in Ethiopia. Anyway, I have some comments and suggestions for the authors.

Comments:

Abstract

#1…see better acronym for Type 2 diabetes

Authors reply: Thank you very much for your suggestion. We used the acronym T2DM to accommodate the word “mellitus” of course many literatures and guidelines also used it. But considering your suggestions and comments we have searched further and T2DM was still found better acronym to type 2 diabetes mellitus.

Introduction

#1… please, about complications includes erectile dysfunction and fracture (see these ref: doi: 10.1002/dmrr.3494; doi: 10.2337/dc18-1965).

Authors reply: We are thankful for your suggestions to include the most important complications of diabetes. By taking your recommendations, we incorporated it in the manuscript (page 3, lines 8-10).

#2… please, update your ref about IDF Atlas ADA standard with the latest.

Authors reply: Thank you for this important reminder to use the latest version. We revised with latest version of IDF diabetes Atlas (page 3, lines 11-16).

#3… after ref 15, 16, about the need to educate patients, you can use ref DOI: 10.1007/s42000-018-0005-9.

Author reply: It is important evidence which further improve the statements and support the arguments. By considering your constructive recommendation, we revised accordingly (page 4, lines 4-5). 

Material and methods

#1…have you used any glucometro in particular?

Author reply: Thank you for the question you raised. Physicians advice their patients how regularly the blood sugar should be monitored either at the health facility (the hospital) by healthcare providers or at home. We have no any data about a particular glucometer used to monitor at home. We used the data from patients’ medical records in the health facility, which is recorded and measured in the hospital setting. 

Results

#1…have you data about HbA1c?

Author reply: Thank you for your comments and questions to give clarification about HbA1c. As we mentioned in the limitation section of the current study, HbA1c was not used to determine glycemic level since it is not available in the study setting and included subjects. Even though it was not consistent, a limited number of subjects had monitored using HbA1c (they have been used it occasionally in private healthcare facilities for some patients who have appointments with their private endocrinologists). It may present for one time monitoring then missed for the next and the rest follow-up periods. Therefore, we examined glycemic control using fasting blood glucose and we have no data about HbA1c.

#2…have you data about the different type of anti hyperglycemic drugs used?

Author reply: As we mentioned in the data collection instruments and procedures section (page 6, lines 20-22), in this study patients with type 2 diabetes had treated by Metformin with or without glibenclamide in pre-period of initiation and then insulin (NPH or premixed) was initiated. After insulin initiation, insulin based therapy was continued either alone or with a single or the two oral agents (metformin and glibenclamide). We also provided in the result section of medication patterns and respective proportion of patients (page 10, lines 1-4) in the 2-year follow-up periods (look at S1 fig.). Therefore, the data about anti hyperglycaemic drugs used is available there. 

 Discussion

#1…about appropriate management, could be important to emphasize the role of educational therapy (see ref. DOI: 10.1007/s42000-018-0005-9) 

Author reply: Thank you very much for this important suggestion. By considering your feedback, we revised the manuscript by incorporating the recommended evidence (page, 13 lines 24-27). 

Reviewer #3

In the current article the authors investigated the effectiveness of taking insulin with or without oral agents among patients with type 2 diabetes mellitus (T2DM) in the resource limited-countries like Ethiopia. Thus, in analysis of retrospective medical records of patients with T2DM who initiated insulin due to uncontrolled hyperglycemia of 424 enrolled patients with T2DM, there was significant deference in FPG before and after insulin initiation (P < 0.001), and a declining trend of blood glucose during a 1-year follow-up period of post-initiation was observed. However, majority of the participants did not achieve target glucose levels. Participants who had higher FPG and systolic blood pressure (SBP) before insulin initiation were found more likely to have poor glycemic control after insulin initiation. Similarly, patients who received atorvastatin than simvastatin were found to have poor glycemic control in the post-period of initiation (P =0.04), whereas premixed insulin was associated with a lower likelihood of poor glycemic control than neutral protamine Hagedorn (NPH) insulin (P < 0.001).

Therefore, they concluded that following insulin initiation, a significant change in glycemic level and declining trend of FPG during a 1-year post-initiation period was established. However, the majority of patients were under poor level of glycemic control. Appropriate management focusing on predictors of glycemic control would be of great benefit to achieve glycemic control.

The article looks as other just published articles on the same theme. However, my question is: what is new from the current article? What is the current article adding to the current literature on the topic of glycemic control in the diagnosis and treatment of the patients with type diabetes? It is well known that the insulin therapy could be used with higher success rate to achieve the best glycemic control in patients with type 2 diabetes. In my opinion we cannot publish this article

Author reply: Dear reviewer, we appreciate your time to review our manuscript. Despite we share some of your concern, we hope that the finding from the current paper showed the real world practices in the low-income counties like Ethiopia which there no prior evidence in such settings. Furthermore, we also believe that the findings were not duplication of a published literature everywhere as trials and guidelines recommendations vary in different settings due to different reasons. 

Though guidelines and trails have acknowledged that initiation of insulin can improve glycemic level of patients with type 2 diabetes but as evidences revealed, it is not easy to achieve the target levels of glycemic control in the real practice settings. Level of glycemic achievement is also different across different study settings due to various reasons from patient’s side or related to management and monitoring practices, and the setting itself. The problem requires real world evidence in particular with low income countries.

Most importantly, in the current paper we had examined factors associated with glycemic control positively or negatively. As literatures showed there are different factors of glycemic control across the globe, which are ranged from sociodemographic characteristics up to clinical and management related variables. So, exploring these factors with real world data could be important to tailor evidence based intervention accordingly. Particularly, in Ethiopia, which is one of the largest diabetes population countries in the sab-Saharan Africa, determining level of glycemic control and analysing the factors of poor glycemic control in new insulin-initiated patients with type 2 diabetes who failed to achieve glucose targets by OADs alone has a lot of implications. The study was carried out using real world data from the tertiary setting by incorporating all available and necessary information from the patients’ side and management practice of the setting. As a result, we have found important contributory factors associated with glycemic control. It can help to intervene the management practice accordingly and a benchmark for further investigation in the study setting. Generally, the current research highlighted the general image of glycemic level and potential factors of glycemic control in insulin initiated patients with type 2 diabetes in the study area, which are important to consider in managing of insulin initiated patients with type 2 diabetes.

---

## [Decision Letter · Decision Letter 1]

4 May 2022

Glycemic control in newly insulin initiated patients with type 2 diabetes mellitus: A retrospective follow-up study at a university hospital in Ethiopia

PONE-D-22-08211R1

Dear Dr. Ashenafi Kibret Sendekie

We’re pleased to inform you that your manuscript has been judged scientifically suitable for publication and will be formally accepted for publication once it meets all outstanding technical requirements.

Kind regards,

Ferdinando Carlo Sasso, PhD, MD

Academic Editor

PLOS ONE

Additional Editor Comments (optional):

Reviewers' comments:

Reviewer's Responses to Questions

**Comments to the Author**

1. If the authors have adequately addressed your comments raised in a previous round of review and you feel that this manuscript is now acceptable for publication, you may indicate that here to bypass the “Comments to the Author” section, enter your conflict of interest statement in the “Confidential to Editor” section, and submit your "Accept" recommendation.

Reviewer #1: All comments have been addressed

Reviewer #2: All comments have been addressed

2. Is the manuscript technically sound, and do the data support the conclusions?

Reviewer #1: Yes

Reviewer #2: Yes

3. Has the statistical analysis been performed appropriately and rigorously? 

Reviewer #1: Yes

Reviewer #2: Yes

4. Have the authors made all data underlying the findings in their manuscript fully available?

Reviewer #1: Yes

Reviewer #2: Yes

5. Is the manuscript presented in an intelligible fashion and written in standard English?

Reviewer #1: Yes

Reviewer #2: Yes

6. Review Comments to the Author

Reviewer #1: The author answered appropriately to all my queries. The manuscript has much improved and, in my opinion, can be further processed for publication.

Reviewer #2: Please, insert all the part not present in the manuscript, underlined by reviewers (as the lack of evaluatoion of HbA1c or data of a type of glucometer) in the text.

7. PLOS authors have the option to publish the peer review history of their article (what does this mean?). If published, this will include your full peer review and any attached files.

Reviewer #1: No

Reviewer #2: No

---

## [Editor Report · Acceptance letter]

18 May 2022

PONE-D-22-08211R1 

Glycemic control in newly insulin-initiated patients with type 2 diabetes mellitus: A retrospective follow-up study at a university hospital in Ethiopia  

Dear Dr. Sendekie:

I'm pleased to inform you that your manuscript has been deemed suitable for publication in PLOS ONE. Congratulations! Your manuscript is now with our production department. 

Kind regards, 

on behalf of

Professor Ferdinando Carlo Sasso 

Academic Editor

PLOS ONE